# Coupling between Nitrification and Denitrification as well as Its Effect on Phosphorus Release in Sediments of Chinese Shallow Lakes

**Yao Zhang [1,2], Chunlei Song [1,*], Zijun Zhou [3], Xiuyun Cao [1] and Yiyong Zhou [1]**

1   Key Laboratory of Algal Biology, State Key Laboratory of Freshwater Ecology and Biotechnology, Institute of Hydrobiology, the Chinese Academy of Sciences, 7# Donghu South Road, Wuhan 430072, China
2   Center for Environment and Health in Water Source Area of South-to-North Water Diversion, School of Public Health and Management, Hubei University of Medicine, Shiyan 442000, China
3   Yellow River Water Resources Protection Institution, Zhengzhou 450004, China
*   Correspondence: clsong@ihb.ac.cn; Tel.: +86 27 68780221; Fax: +86 27 68780709

**Abstract:** The coupling of nitrification and denitrification has attracted wide attention since it plays an important role in mitigating eutrophication in aquatic ecosystems. However, the underlying mechanism is largely unknown. In order to study the coupling relationship between nitrification and denitrification, as well as its effect on phosphorus release, nutrient levels, functional gene abundance and potential rates involved in nitrification and denitrification were analyzed in three shallow urban lakes with different nutrient status. Trophic level was found positively related to not only copy numbers of functional genes of nitrosomonas and denitrifiers, but also the potential nitrification and denitrification rates. In addition, the concentrations of different forms of phosphorus showed a positive correlation with the number of nitrosomonas and denitrifiers, as well as potential nitrification and denitrification rates. Furthermore, the number of functional genes of nitrosomonas exhibited positive linear correlations with functional genes and rate of denitrification. These facts suggested that an increase in phosphorus concentration might have promoted the coupling of nitrification and denitrification by increasing their functional genes. Strong nitrification–denitrification fueled the nitrogen removal from the system, and accelerated the phosphorus release due to the anaerobic state caused by organic matter decomposition and nitrification. Moreover, dissolved organic nitrogen was also released into the water column during this process, which was favorable for balancing the nitrogen and phosphorus ratio. In conclusion, the close coupling between nitrification and denitrification mediated by nitrifier denitrification had an important effect on the cycling mode of nitrogen and phosphorus.

**Keywords:** denitrification; nitrification; phosphorus release; sediment

## 1. Introduction

The biogeochemical cycle of nitrogen in aquatic ecosystems has consistently been a significant issue [1]. Within this cycle, nitrification and denitrification are key steps and are mainly catalyzed by microbes [2,3]. Nitrification, the microbial oxidation of ammonia to nitrite and nitrate, keeps N in the water ecosystem [4,5]. Denitrification, meanwhile, removes N from the water ecosystem by converting nitrate into gaseous nitrogen [6]. Nitrification take place in the aerobic layer while denitrification is conducted in low-oxygen or anaerobic conditions, and the production of $N_2O$ is found to be highest at the interfaces between the two areas [7]. Coupled nitrification and

denitrification (CND) plays an important role in this process. $NO_2^-$ or $NO_3^-$ produced during nitrification can be quickly transferred to the anaerobic zone and utilized by denitrifying bacteria [8].

The CND has received wide attention in recent years, even though the mechanism has remained unclear. Ammonia oxidation, the first and rate-limiting step of nitrification, is catalyzed by ammonia-oxidizing archaea and bacteria. The amoA gene, which encodes the catalytic subunit of ammonia monooxygenase, has been widely used as a functional marker to analyze their communities [9]. Denitrification consists of four reaction steps and is mainly catalyzed by four enzymes: nitrate reductase, nitrite reductase, nitric-oxide reductase and nitrous oxide reductase [10]. Nitrite reductase (*Nir: nirS or nirK*) and nitrous oxide reductase (*Nos: nosZ*) are mainly used for the molecular ecology of denitrifying bacteria [11]. The coupling of nitrification and denitrification, mediated by respective microorganisms, could be reflected by the abundance of the related bacteria [12].

The CND accelerates the N removal rate and thus mitigates eutrophication in aquatic ecosystems [13]. Although the CND is known to play a critical role in removing excessive N from aquatic ecosystems, limited information about the mechanism in this combined process is known. Studying the relationship between the nitrifying and denitrifying microorganisms that are involved helps toward a better understanding of the N cycle in aquatic ecosystems [14]. Except for the significant effect on the N cycle, the CND also demonstrates superior P release [15]. This is mainly because the anaerobic state accelerates the P release [16]. N removal and P release can be achieved simultaneously [17].

In this study, three shallow lakes with different nutrient gradients in Wuhan city were selected to analyze nutrient levels, functional genes abundance and potential rates involved in nitrification and denitrification. The aim was to examine the relationship between nitrogen and phosphorus levels, nitrifying and denitrifying bacteria abundance and their potential rate. We hope to verify the hypothesis that the abundance of nitrifying and denitrifying bacteria determined the potential rate of nitrification and denitrification as well as their coupling relationship, which further stimulated the nitrate removal and phosphorus release. The purpose of this study is to 1) examine the relationships between trophic level, functional gene abundance and potential rates involved in nitrification and denitrification; 2) test whether the coupling between nitrification and denitrification exists and determine its mechanism; 3) estimate the effect of the coupling of nitrification and denitrification on N and P release.

## 2. Materials and Methods

### 2.1. Study Sites and Sample Collection

In this study, samples from 16 sites at four representative lakes or zones (Lake Houguan, eutrophic, macrophyte coverage; Lake Tangxun, strongly hypertrophic, aquaculture and anoxia; South of Lake Qingling, lightly hypertrophic, macrophyte coverage; the North of Lake Qingling, strongly hypertophic, aquaculture and pollutant discharge) were collected on May 5, 2015 (Figure 1). The conditions on this date were sunny, the air temperature was about 28 °C, the water temperature was about 27 °C.

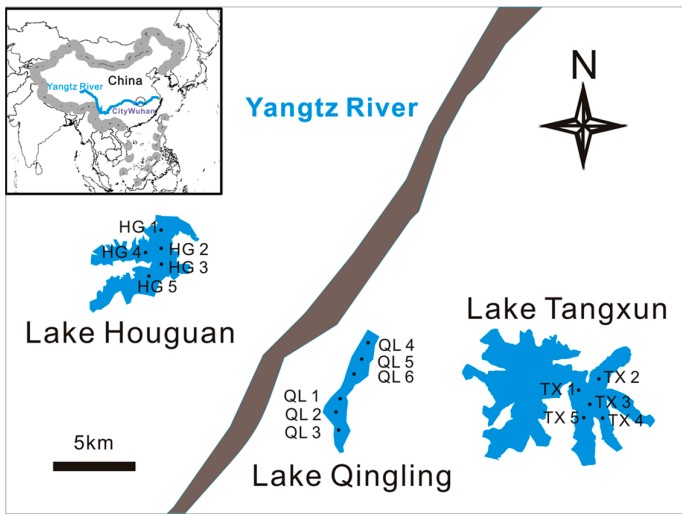

**Figure 1.** Distribution map of sampling lakes and sites in Wuhan city (Lake Houguan: HG1–HG5; Lake Tangxun: TX1–TX5; South of Lake Qingling: QL1–QL3; North of Lake Qingling: QL4–QL6).

Mixed water samples were collected with a Friedinger sampler for nutrient level and extracellular enzyme activity (EEA) level analysis. The transparency (Trans) was measured by a Secchi disk, a circular disk with alternating black and white quadrants that is lowered into the water until it disappears from view [18]. The basic data from each sampling site are shown in Table 1. Surface sediments (0–10 cm) were sampled using a Peterson grab sampler. All the samples were immediately stored in cooling boxes for transportation to the laboratory. Water samples were processed the same day, sediment samples were stored at 4 °C in the dark for up to four days before analysis.

**Table 1.** Water depth, transparency, dissolved oxygen (DO), pH, chlorophyll *a* (Chl *a*), total phosphorus (TP) and trophic state index (TSI) data of the studied lakes.

| | Water Depth (m) | Transparency (m) | Chl *a* ($\mu g\ L^{-1}$) | DO ($mg\ L^{-1}$) | pH | TP ($\mu g\ L^{-1}$) | TSI |
|---|---|---|---|---|---|---|---|
| HG1 | 1.10 | 0.25 | 27.62 | 7.80 | 8.30 | 32.26 | 66.69 |
| HG2 | 1.20 | 0.30 | 20.41 | 7.90 | 8.30 | 21.98 | 63.41 |
| HG3 | 1.25 | 0.50 | 18.69 | 8.80 | 8.78 | 20.05 | 60.54 |
| HG4 | 1.30 | 0.65 | 16.92 | 8.30 | 8.67 | 20.69 | 58.96 |
| HG5 | 1.30 | 0.75 | 13.28 | 6.70 | 8.40 | 17.47 | 56.66 |
| TX1 | 2.30 | 0.25 | 487.36 | 15.24 | 9.83 | 290.74 | 87.07 |

| | | | | | | |
|---|---|---|---|---|---|---|
| TX2 | 2.33 | 0.35 | 147.56 | 15.94 | 9.72 | 248.94 | 78.93 |
| TX3 | 2.41 | 0.35 | 92.07 | 14.26 | 9.72 | 167.29 | 75.50 |
| TX4 | 2.40 | 0.30 | 79.75 | 14.80 | 9.76 | 29.05 | 71.28 |
| TX5 | 2.43 | 0.20 | 194.45 | 17.70 | 9.96 | 124.21 | 81.16 |
| QL1 | 1.50 | 0.50 | 65.94 | 2.74 | 7.68 | 160.86 | 72.11 |
| QL2 | 1.50 | 0.45 | 48.48 | 5.74 | 7.90 | 162.79 | 70.96 |
| QL3 | 1.30 | 0.30 | 62.40 | 6.78 | 8.21 | 103.63 | 72.97 |
| QL4 | 1.50 | 0.30 | 222.64 | 8.50 | 8.70 | 868.76 | 84.71 |
| QL5 | 1.50 | 0.30 | 338.81 | 9.72 | 8.90 | 902.84 | 87.03 |
| QL6 | 1.50 | 0.30 | 296.47 | 9.75 | 8.93 | 972.92 | 86.50 |

## 2.2. Chemical Analysis of Water and Sediment Samples

Water samples were filtered through a 0.45 μm membrane filter for analysis of soluble nutrients. All of the nutrient measurements, including ammonium ($NH_4^+$-N), nitrate ($NO_3^-$-N), nitrite ($NO_2^-$-N), dissolved total nitrogen (DTN), total nitrogen (TN), soluble reactive phosphorus (SRP), dissolved total phosphorus (DTP) and total phosphorus (TP), were measured based on national standards (APHA 2012). Dissolved organic nitrogen (DON) was calculated as follows: DON = DTN-DIN. Chlorophyll *a* (Chl *a*) was extracted from GF/C filters (Whatman, USA) with 95% ethanol and measured at wavelengths of 665 nm and 750 nm [19]. Trophic state index (TSI) was calculated using three limological parameters, namely, Chl *a* (μg L$^{-1}$), Secchi disk transparency (Trans) (m) and TP (μg L$^{-1}$), according to Carlson [20].

The extracellular enzyme activity (EEA) levels, including alkaline phosphatase activity (APA), β-D-glucosidase activity (GLU) and leucine aminopeptidase activity (LAP), were determined fluorometrically, according to Boetius and Lochte [21]. Briefly, the methylumbelliferone (MUF)-labeled substrates MUF-phosphate, MUF-Glu (β-D-glucopyranoside) and MCA-Leu (L-leucine-4-methylcoumarinyl-7-amid HCl) were used as their respective substrates.

Sediment P was divided into iron-bound P (Fe(OOH)~P), calcium-bound P (CaCO$_3$~P), acid-soluble organic P (ASOP) and hot NaOH-extractable organic P ($P_{alk}$), according to Golterman [22]. Different forms of P were extracted sequentially and determined as SRP concentrations by the molybdate blue method, according to Murphy and Riley [23].

## 2.3. Determination of Potential Denitrification Rate and Potential Nitrification Rate of Sediment

The sediment potential denitrification rate (PDR) was measured by denitrifying enzyme activity assay, according to Jha and Minagwa [24]. Briefly, 5 g of sediment was moved into a special tailor-made glass bottle, then 20 mL of denitrifying enzyme activity solution (7 mM KNO$_3$, 3 mM glucose and 5 mM chloramphenicol) was added. The air in the bottle was purged by a continuous pumping

of helium, then acetylene was added to reach a final concentration of 10%. Samples were placed on an orbital incubator and shaken (125 r/min) in the dark at 25 °C. Gas samples were taken out at 0, 0.5, 1, 1.5, and 2 h for testing of $N_2O$ concentration using a gas chromatograph. The PDR was calculated as a line of best fit curve for the $N_2O$ concentration as a function of time.

The sediment potential nitrification rate (PNR) was measured according to the shaken-slurry method [25]. Briefly, 5.0 g of sediment was moved into a 250 mL sterile Erlenmeyer flask. Then, 100 mL of phosphate buffer (1 mL, pH 7.4) and 0.5 mL of $(NH_4)_2SO_4$ (0.25 M) were added. Samples were incubated on an orbital shaker (180 rpm) at 25 °C for 24 h. A total of 5 mL of slurry was taken out at 1, 4, 10, 16 and 24 h, then the slurry was centrifuged at 4000 rpm for 5 min and filtered through glass microfiber filters (Whatman) for $NO_3^--N$ measurement. The PNR was calculated as the $NO_3^--N$ production per unit of time.

## 2.4. DNA Extraction and qPCR

DNA was extracted from a 0.3 g fresh sediment sample by an UltraClean Soil DNA Isolation Kit (MoBio Laboratories, Carlsbad, CA). Quantitative polymerase chain reaction (qPCR) assays were performed using respective primers. Briefly, the primers were *nir*S1F (CCT AYT GGC CGG CRC ART) and *nir*S3R (GCC GCC GTC RTG VAG GAA) [26] for the *nir*S gene, and reactions were performed with 10 ng of template DNA and 0.1 μM concentrations of each primer in a total volume of 20 μL. The cycling conditions were: 95 °C for 30 s, followed by 30 cycles of 5 s at 95 °C, 30 s at 57 °C and 40 s at 72 °C, and then, 7 min at 72 °C. F1aCu(ATC ATG GTS CTG CCG CG) and R3Cu(GCC TCG ATC AGR TTG TGG TT) [27] were used for the *nir*K gene, and reactions were performed with 10 ng of template DNA and 0.1 μM concentrations of each primer in a total volume of 20 μL. The cycling conditions were: 95 °C for 30 s, followed by 30 cycles of 5 s at 95 °C, 40 s at 55 °C and 50 s at 72 °C, and then 7 min at 72 °C.

*nos*Z1F(WCS YTG TTC MTC GAC AGC CAG) and *nos*Z1R(ATG TCG ATC ARC TGV KCR TTY TC) [28] were used for the *nos*Z1 gene, and reactions were performed with 10 ng of template DNA and 0.1 μM concentrations of each primer in a total volume of 20 μL. The cycling conditions were: 95 °C for 30 s, followed by 30 cycles of 5 s at 95 °C, 50 s at 58 °C and 30 s at 72 °C, then 7 min at 72 °C. *nos*Z2F (CGC RAC GGC AAS AAG GTS MSS GT) and *nos*Z2R (CAK RTG CAK SGC RGT TCA GAA) [28] were used for the *nos*Z2 gene, and reactions were performed with 10 ng of template DNA and 0.1 μM concentrations of each primer in a total volume of 20 μL. The cycling conditions were: 95 °C for 30 s, followed by 30 cycles of 5 s at 95 °C, 50 s at 58 °C and 30 s at 72 °C, then 7 min at 72 °C.

ArchamoA-1F (STA ATG GTC TGG CTT AGA CG) and ArchamoA-2R (GCG GCC ATC CAT CTG TAT GT) [4] were used for ammonia-oxidizing archaea (AOA) and the amoA gene, and reactions were performed with 10 ng of template DNA and 0.1 μM concentrations of each primer in a total volume of 20 μL. The cycling conditions were: 95 °C for 30 s, followed by 35 cycles of 5 s at 94 °C, 30 s at 53 °C and 30 s at 72 °C, then 8 min at 72 °C. amoA-1F(GGG GTT TCT ACT GGT GGT) and amoA-2R(CCC CTC KGS AAA GCC TTC TC) [29] were used for ammonia-oxidizing bacteria (AOB) and the amoA gene, and reactions were performed with 10 ng of template DNA and 0.1 μM concentrations of each primer in a total volume of 20 μL. The cycling conditions were: 95 °C for 30 s, followed by 35 cycles of 5 s at 94 °C, 30 s at 55 °C and 30 s at 72 °C, then 8 min at 72 °C.

## 2.5. Statistical Analysis

The Pearson's test was performed using the SPSS statistical software (version 18.0, SPSS, Chicago, IL, USA), with a value of 0.05 or 0.01 selected for significance. Data were transformed and tested for normality before correlation analysis.

## 3. Results

The TSI results of sampling sites ranged from 56.66 to 87.07 (Table 1). The lowest and highest P levels in the water were observed in Lake Houguan and North of Lake Qingling (QL4-6), respectively. SRP concentration was highest in North of Lake Qingling (Figure 2a). The main form of P in sediment was Fe(OOH)~P, observed in North of Lake Qingling (Figure 2b). The N content exhibited significantly higher levels in Lake Tangxun and North of Lake Qingling. The lowest N level was observed in Lake Houguan. The main N form was DON in Lake Tangxun and Lake Qingling. The different N form was at an equilibrium level in Lake Houguan (Figure 2c). Lake Tangxun showed the highest EEA level especially APA (Figure 2d). The highest PDR and PNR value were recorded in North of Lake Qingling, which was followed by Lake Tangxun. The relatively low PDR and PNR value were found in Lake Houguan and South of Lake Qingling (Figure 2e).

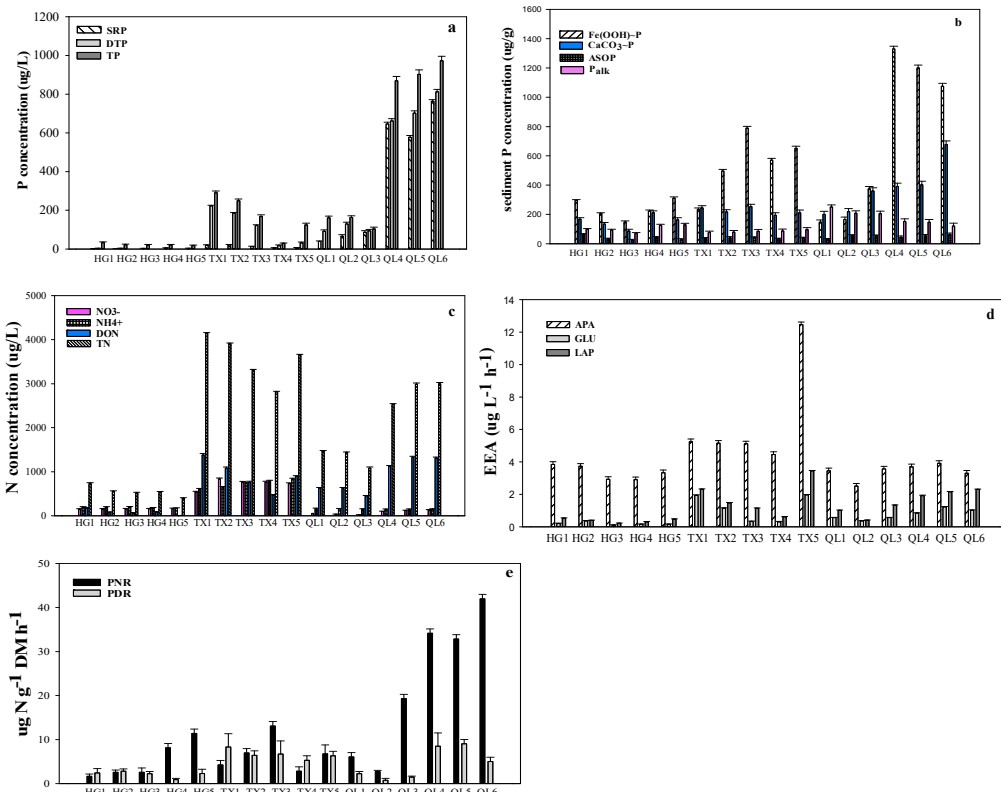

**Figure 2.** Comparison of P (**a** and **b**), N (**c**), extracellular enzyme activity (EEA) (**d**), and potential nitrification rate (PNR)/potential denitrification rate (PDR) (**e**) at different sampling sites.

The abundance of functional genes showed the distinct difference in different lakes. For example, the AOB gene was significantly higher in Lake Qingling, the AOA, *nir*S and *nir*K gene showed a low level in all lakes, while the *nos*Z1 and *nos*Z2 genes were significantly higher in Lake Qingling and Lake Tangxun (Table 2). The AOA-amoA and AOB-amoA gene copy numbers ranged from $9.57 \times 10^6$ to $6.90 \times 10^7$ and from $6.90 \times 10^6$ to $1.94 \times 10^9$ $g^{-1}$ for dry sediment, respectively (Table 2). The *nir*K and *nir*S gene copy numbers ranged from $7.01 \times 10^5$ to $1.48 \times 10^8$ and from $8.93 \times 10^6$ to $4.83E \times 10^7$ $g^{-1}$ for dry sediment, respectively (Table 2). The *nos*Z1 and *nos*Z2 gene copy numbers ranged from $2.22 \times 10^7$ to $9.44 \times 10^8$ and from $8.28 \times 10^6$ to $8.35 \times 10^8$ $g^{-1}$ for dry sediment, respectively (Table 2).

**Table 2.** Copy numbers of ammonia-oxidizing archaea (AOA)-amoA, ammonia-oxidizing bacteria (AOB)-amoA, *nir*K, *nir*S, *nos*Z1 and *nos*Z2 at different sampling sites.

| title | AOA-amoA | AOB-amoA | *nir*K | *nir*S | *nos*Z1 | *nos*Z2 |
|---|---|---|---|---|---|---|
| HG1 | $1.11 \times 10^7$ | $1.89 \times 10^7$ | $2.27 \times 10^6$ | $1.02 \times 10^7$ | $1.46 \times 10^8$ | $1.24 \times 10^8$ |
| HG2 | $9.57 \times 10^6$ | $1.32 \times 10^7$ | $7.01 \times 10^5$ | $8.93 \times 10^6$ | $1.17 \times 10^8$ | $1.22 \times 10^8$ |
| HG3 | $1.67 \times 10^7$ | $2.00 \times 10^8$ | $1.25 \times 10^6$ | $1.14 \times 10^7$ | $4.38 \times 10^8$ | $3.09 \times 10^8$ |
| HG4 | $1.62 \times 10^7$ | $6.90 \times 10^6$ | $1.60 \times 10^6$ | $1.41 \times 10^7$ | $2.22 \times 10^7$ | $8.28 \times 10^6$ |
| HG5 | $1.04 \times 10^7$ | $9.46 \times 10^6$ | $2.67 \times 10^6$ | $9.62 \times 10^6$ | $4.45 \times 10^7$ | $1.24 \times 10^7$ |
| TX1 | $1.49 \times 10^7$ | $4.68 \times 10^7$ | $1.39 \times 10^6$ | $1.31 \times 10^7$ | $3.90 \times 10^8$ | $8.35 \times 10^8$ |
| TX2 | $1.11 \times 10^7$ | $2.31 \times 10^7$ | $3.18 \times 10^6$ | $1.02 \times 10^7$ | $4.27 \times 10^8$ | $3.88 \times 10^8$ |
| TX3 | $1.99 \times 10^7$ | $5.41 \times 10^7$ | $3.58 \times 10^6$ | $1.67 \times 10^7$ | $6.00 \times 10^8$ | $5.10 \times 10^8$ |
| TX4 | $3.09 \times 10^7$ | $2.67 \times 10^8$ | $3.43 \times 10^7$ | $2.45 \times 10^7$ | $3.63 \times 10^8$ | $7.48 \times 10^8$ |
| TX5 | $1.92 \times 10^7$ | $2.87 \times 10^7$ | $2.46 \times 10^7$ | $1.62 \times 10^7$ | $2.56 \times 10^8$ | $4.12 \times 10^8$ |
| QL1 | $4.83 \times 10^7$ | $6.68 \times 10^8$ | $2.03 \times 10^6$ | $3.56 \times 10^7$ | $5.85 \times 10^8$ | $3.64 \times 10^8$ |
| QL2 | $2.20 \times 10^7$ | $1.39 \times 10^8$ | $1.46 \times 10^8$ | $1.81 \times 10^7$ | $6.21 \times 10^8$ | $2.68 \times 10^8$ |
| QL3 | $4.52 \times 10^7$ | $2.75 \times 10^8$ | $1.48 \times 10^8$ | $3.36 \times 10^7$ | $1.74 \times 10^8$ | $7.65 \times 10^8$ |
| QL4 | $6.90 \times 10^7$ | $1.75 \times 10^9$ | $8.46 \times 10^7$ | $4.83 \times 10^7$ | $3.06 \times 10^8$ | $7.60 \times 10^8$ |
| QL5 | $5.15 \times 10^7$ | $1.94 \times 10^9$ | $2.82 \times 10^7$ | $3.78 \times 10^7$ | $9.44 \times 10^8$ | $7.30 \times 10^8$ |
| QL6 | $6.48 \times 10^7$ | $1.06 \times 10^9$ | $8.15 \times 10^7$ | $4.61 \times 10^7$ | $3.00 \times 10^8$ | $5.44 \times 10^8$ |

Significantly positive relationships existed between P concentrations, including DTP, SRP, Fe(OOH)~P, $CaCO_3$~P, ASOP, and $P_{alk}$, and copy numbers of different gene types of denitrifying bacteria, including *nir*S , *nir*K, *nos*Z1 and *nos*Z2 (Table 3 and Figure 3a, 3b, 3e, 3f). Copy numbers of *nir*S and *nos*Z2 showed significant positive correlations with LAP (Table 3 and Figure 3g). Furthermore, the TN concentration showed significant positive correlations with the copy numbers of *nos*Z1 and *nos*Z2 (Table 3 and Figure 3h).

**Table 3.** Pearson's correlation coefficients between nutrient forms and copy numbers of AOA/AOB as well as denitrifiers.

| | AOA | AOB | *nir*K | *nir*S | *nos*Z1 | *nos*Z2 | PDR | PNR |
|---|---|---|---|---|---|---|---|---|
| SRP | 0.845** | 0.795** | 0.675** | 0.844** | 0.598* | 0.727** | 0.667** | 0.301 |

| | | | | | | | | |
|---|---|---|---|---|---|---|---|---|
| DTP | 0.696** | 0.670** | 0.545* | 0.695** | 0.644** | 0.653** | 0.674** | 0.546* |
| Fe(OOH)~P | 0.546* | 0.449 | 0.536* | 0.548* | 0.253 | 0.415 | 0.739** | 0.717** |
| Ca(OOH)~P | 0.780** | 0.574* | 0.682** | 0.780** | 0.218 | 0.401 | 0.827** | 0.352 |
| ASOP | 0.311 | 0.230 | 0.562* | 0.370 | 0.108 | 0.200 | 0.318 | −0.032 |
| P$_{alk}$ | 0.595* | 0.427 | 0.492 | 0.592* | 0.027 | −0.025 | 0.345 | −0.445 |
| APA | −0.237 | −0.199 | −0.010 | −0.116 | 0.169 | 0.317 | 0.004 | 0.604* |
| GLU | 0.315 | 0.343 | 0.357 | 0.410 | 0.470 | 0.630** | 0.414 | 0.652** |
| LAP | 0.523* | 0.405 | 0.422 | 0.524* | 0.422 | 0.615* | 0.604 | 0.726** |
| NO$_3^-$-N | −0.386 | −0.377 | −0.367 | −0.405 | 0.074 | 0.081 | −0.183 | 0.621* |
| NO$_2^-$-N | −0.383 | −0.268 | −0.009 | −0.31 | 0.155 | 0.148 | −0.079 | 0.641* |
| NH$_4^+$-N | −0.297 | −0.319 | −0.172 | −0.318 | 0.233 | 0.294 | −0.274 | 0.506* |
| TN | 0.352 | 0.421 | 0.403 | 0.44 | 0.672** | 0.750** | 0.347 | 0.742** |
| TSI | 0.596* | 0.588* | 0.494 | 0.597* | 0.660** | 0.795** | 0.501* | 0.722** |

Significance at **$\alpha$ = 0.01 level, *$\alpha$ = 0.05 level; n = 16.

Data were transformed and tested for normality before correlation analysis.

Abbreviations of nutrient forms: SRP, soluble reactive phosphorus; DTP, dissolved total phosphorus; Fe(OOH)~P, iron-bound P; Ca(OOH)~P, calcium-bound P; ASOP, acid-soluble organic P; P$_{ALK}$, hot NaOH-extractable organic P; APA, alkaline phosphatase activity; GLU, β-D-glucosidase activity; LAP, leucine aminopeptidase activity; NO$_3^-$-N, nitrate; NO$_2^-$-N, nitrite; NH$_4^+$-N, ammonium; TN, total

nitrogen.

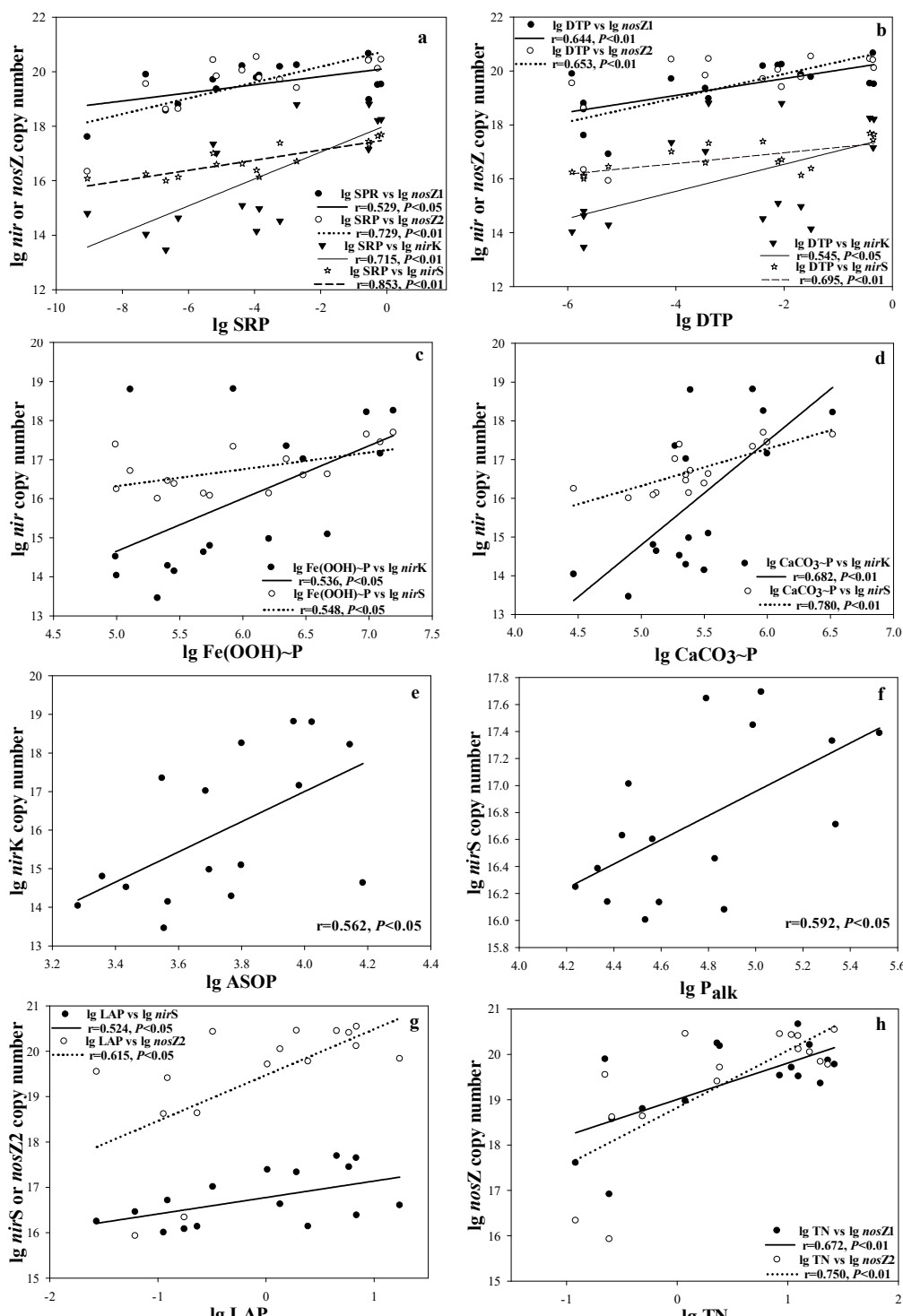

**Figure 3.** Scatter plots of the correlations between copy number of denitrifiers and SRP (**a**), DTP (**b**), Fe(OOH)~P (c), CaCO₃~P (d), ASOP (e), P$_{alk}$ (f), LAP (g), TN (h) (n = 16).

Meanwhile, PDR and PNR showed significant and positive relationships with P concentrations (Figure 4a, 4b) as well as TSI (Figure 4d). Additionally, PDR was significantly and positively related to the copy numbers of the *nir*S and *nir*K genes (Figure 4c). AOA and AOB abundance exhibited

significant    positive    relationships    with    SRP,    DTP,    CaCO₃~P    and    TSI    (Table    3).

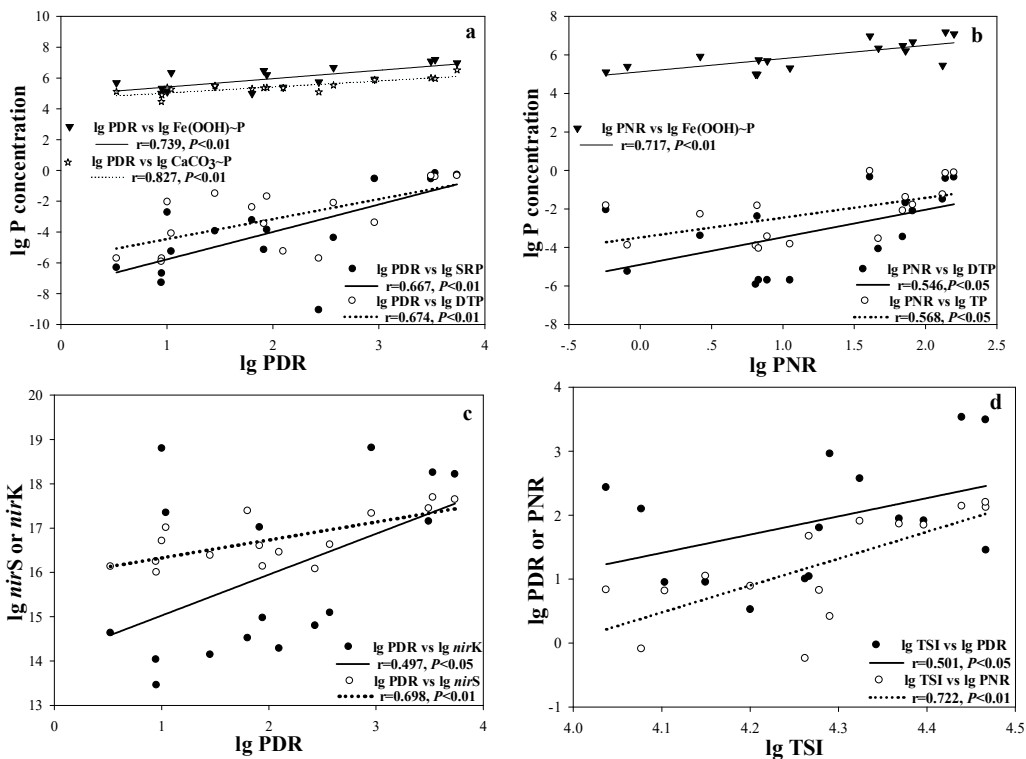

**Figure 4.** Scatter plots of the correlations between P concentration and PDR (**a**), PNR (**b**); PDR and copy numbers of *nir*S and *nir*K (**c**); TSI and PDR/PNR (**d**) (n = 16).

Noticeably, the abundance of AOA and AOB showed positive linear correlations, not only with denitrifying bacteria encoding the different gene types (Figure 5a and b) but also with PDR (Figure5c). In turn, PNR was significantly and positively related to copy numbers of the *nos*Z2 gene (Figure 5d).

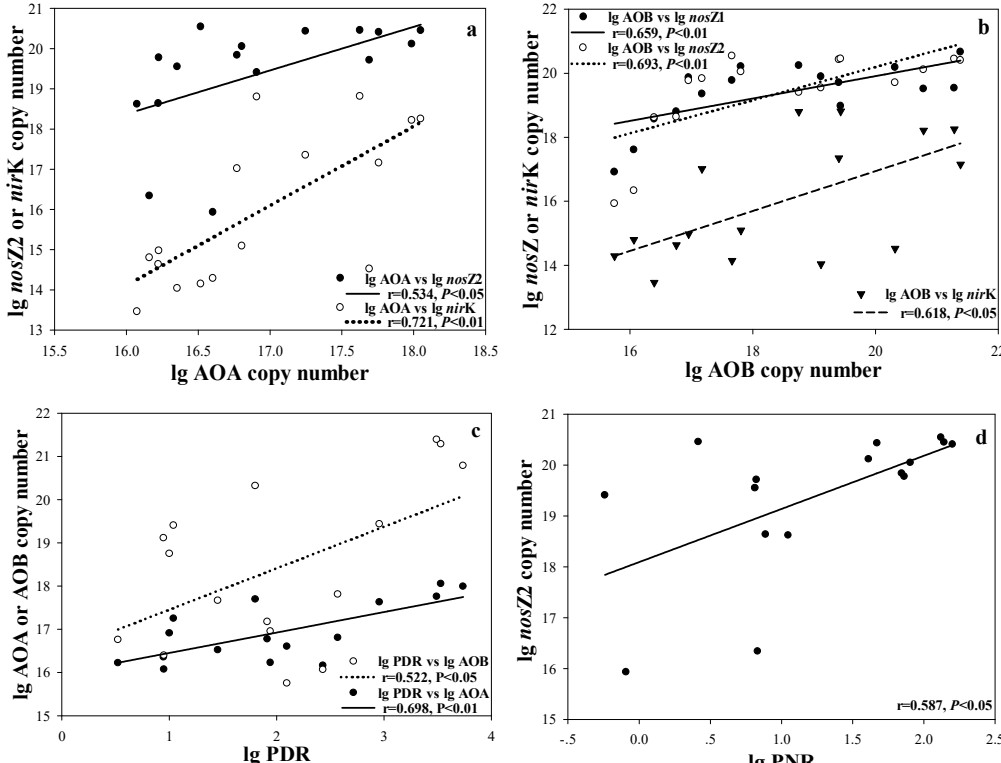

**Figure 5.** Scatter plots of the correlations between AOA, AOB and denitrifying bacteria (**a**, **b**); PDR, PNR and copy numbers of functional genes (**c**, **d**) (n = 16).

## 3. Discussion

Copy numbers of the AOB-*amo*A gene at most sampling sites were higher than those of the AOA-*amo*A gene. Similarly, in a constructed wetland, AOB (ranging from $6.76 \times 10^5$ to $6.01 \times 10^7$ per gram of dry sediment) were much more abundant than AOA (ranging from below detection limit to $9.62 \times 10^6$ per gram of dry sediment) [30]. Many previous studies have revealed considerable changes in AOA and AOB abundance in different ecological environments [31]. In this study, the copy numbers of the *nir*S gene were higher than those of the *nir*K gene at 10 sampling sites, and the copy numbers of the *nos*Z gene were much higher than those of *nir*S as well as *nir*K. A number of studies have shown different results when comparing the copy numbers of *nir*S and *nir*K genes [2,32]. In line with our study, the nosZ gene generally exhibited higher abundance than *nir*S as well as *nir*K in activated sludge [33].

In this study, we observed a close relationship between the nutritional status of a lake and the abundance as well as activity of nitrosomonas and denitrifiers (Table 3, Figure 3 and Figure 4). These results indicate that the trophic state of lakes determines the microbial activity and functional gene abundance. In Cape Fear River, significant seasonal variation in bacterial *amo*A, *nir*S and *nos*Z gene abundance was largely explained by increases in substrate availability [12]. An improvement in soil fertility (mineral or organic matter fertilization) could increase the abundance of *nir*S and the proportion of potential $N_2O$, and a strong correlation was observed between *nir*S gene abundance and potential $N_2O$ emissions [34]. Additionally, denitrification rates in streams were controlled primarily by sediment organic matter and *nos*Z abundance [35]. In this study, it was found that copy numbers of both *nir*S and *nir*K genes were positively correlated with PDR (Figure 4c), while *nos*Z did not present the same relationship. In eutrophic lakes, the enrichment of total organic carbon and all forms of P in sediments could fuel PDR by shaping community composition and increasing the

abundance of *nir*S-type denitrifiers [36]. In soils, urea addition increased *nir*S and *nir*K gene abundances and N$_2$O emission [37]. A quantification of the functional genes involved in denitrification along with the Spearman's rank correlation matrix revealed that the N$_2$O emission rates correlated with the abundance of *nir*K and *nir*S genes [38]. In short, lake eutrophication aggravation greatly increased the abundance of nitrifying and denitrifying bacteria with specific functional genes (especially *nir*S and *nir*K), which further effectively promoted nitrification and denitrification rates.

Our results also showed a connection between nitrification and denitrification (Figure 5). These results indicated that the coupling of nitrification and denitrification was based on the combination of microbial quantity and function. This coupling can be explained by the nitrifier denitrification mechanism. The first step of nitrification is accomplished by AOB and AOA with the participation of O$_2$, while both AOA and AOB can produce N$_2$O in the form of by-products [39]. Firstly, ammonium hydroxide or its derivatives formed during the conversion of ammonia to nitrite are oxidized and cracked to produce N$_2$O. Secondly, the process of nitrite to NO is catalyzed by the nitrite reductase (*nir*) gene, which is encoded by the *nir*K and *nir*S genes. The NO reductase (*nor*) gene, which further catalyzes the reaction of NO to N$_2$O, is coded by *nor*B. It is noteworthy that all the detected AOB contain *nir*K and *nor*B genes [40]. Besides, AOA can also reveal homologues of *nir*K and *nor*B [40,41]. These two independent processes producing N$_2$O are called nitrifier denitrification [6]. Thus, besides the classical denitrification process, nitrifier denitrification achieving the coupling of nitrification and denitrification might be another important and efficient pathway for nitrogen removal in shallow lakes. Hence, the aggravation of eutrophication accelerates this process and fuels nitrogen removal.

Meanwhile, P concentrations were positively related with the abundance as well as the activity of nitrosomonas and denitrifiers (Figure 3 and Figure 4). These results indicate that high potential nitrification and denitrification rates stimulated the phosphorus release from the sediments. This result was further proved by the higher P concentrations in more eutrophic sites, such as QL4-6 and TX1-2 (Table 1 and Figure 2). It was indicated that the enrichment of organic carbon and nitrogen can result in P release through the anaerobic status caused by organic matter decomposition, suggesting a close coupling between carbon and N and P cycles [42]. Also, a hypothesis about mutual coupling and interplay between N and P cycling was presented, based on N loss due to P accumulation and P release due to anoxia, which resulted from organic matter decomposition in the process of eutrophication [36]. In this study, besides the fact that organic matter decomposition resulted in anoxia, the strong nitrification that consumed a great deal of oxygen should be considered as an important reason for the formation of anoxia, based on the relationship between P and nitrification. In a manipulated experiment, nitrification in sediment was enhanced on the condition of anoxia, which quickly caused a release of dissolved P [43]. This can be explained by the fact that the enhancement of nitrification resulting in anoxia induced SRP release. Conversely, during this process, DON was also released into the water column in considerable quantity in eutrophic lakes (Figure 2c). Ligand exchange was considered as the important mechanism for the desorption of DON in Lake Taihu sediment. Both sulfate and phosphate had a significant influence on the release of DON, which might have an important influence on N cycling in the water column [44]. Even though the mechanism is still not clear, it is proposed that DON release was beneficial for balancing the disequilibrium of N and P ratio in the water column due to SRP release.

## 4. Conclusions

Taken together, compared to eutrophic Lake Houguan, the enrichment of nutrients (P in particular) in sediments of the strongly hypertophic North of Lake Qingling greatly increased nitrifying and denitrifying microbial abundance by an order of magnitude, further accelerating the potential rates of nitrification and denitrification by 6.9 and 3.5 fold respectively. The coupling of these two processes were especially mediated by nitrifier denitrification. Strong nitrification–denitrification in sediments of North of Lake Qingling fueled the nitrogen removal from the system

in terms of 21.9% and 27.0% lower ammonium and nitrate levels than that of Lake Houguan. Furthermore, this coupling also accelerated the P release (1000 times higher SRP value in the water column of North of Lake Qingling, comapred to Lake Houguan) due to the anaerobic state caused by organic matter decomposition and nitrification. During this process, DON was also released into the water column in considerable quantity, which was favorable for balancing the N and P ratio.

**Author Contributions**: Conceptualization, X.C. and Y.Z.; Data curation, X.C.; Funding acquisition, Y.Z., C.S. and Y.Z.; Investigation, Y.Z. and Z.Z.; Methodology, Y.Z.; Software, Y.Z.; Supervision, C.S.; Validation, Y.Z. and Z.Z.; Writing—original draft, Y.Z.; Writing—review and editing, C.S.

**Funding**: This work was supported by the National Key Research and Development Program of China (2018YFD0900701), National Natural Science Foundation of China (41877381; 41573110; 41807409), the Major Science and Technology Program for Water Pollution Control and Treatment (2017ZX07603), State Key Laboratory of Freshwater Ecology and Biotechnology (2019FBZ01), the Guiding Projects of Scientific Research Program of Hubei Education Department (B2018119), and Cultivating Project for Young Scholar at Hubei University of Medicine (2016QDJZR16).

**Acknowledgments:** We thank Jie Hou and Xi Chen for their guidance on experiments.

**Conflict of Interest**: The authors declare that they have no conflict of interest.

## List of Abbreviations.

| | |
|---|---|
| CND. | Coupled nitrification and denitrification. |
| TN. | Total Nitrogen. |
| DTN. | Dissolved Total Nitrogen. |
| DON. | Dissolved Organic Nitrogen. |
| $NH_4^+$-N. | Ammonium (nitrogen). |
| $NO_3^-$-N. | Nitrate (nitrogen). |
| $NO_2^-$-N. | Nitrite (nitrogen). |
| TP. | Total Phosphorus. |
| DTP. | Dissolved Total Phosphorus. |
| SRP. | Soluble Reactive Phosphorus. |
| $Fe(OOH)$~P. | Iron-bound phosphorus. |
| $CaCO_3$~P. | Calcium-bound phosphorus. |
| ASOP. | Acid-Soluble Organic Phosphorus. |
| $P_{alk.}$ | Hot NaOH-extractable organic phosphorus. |
| EEA. | Extracellular Enzyme Activity. |
| APA. | Alkaline Phosphatase Activity. |
| GLU. | β-D-glucosidase activity. |

LAP.            Leucine aminopeptidase activity.

Chl *a*.        Chlorophyll *a*.

TSI.            Trophic State Index.

PDR.           Potential Denitrification Rate.

PNR.           Potential Nitrification Rate.

AOA.           Ammonia-Oxidizing Archaea.

AOB.           Ammonia-Oxidizing Bacteria.

*nir*.          Nitrite reductase.

*nos*.          Nitrous oxide reductase.

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
