# Peer review of "Coupling between Nitrification and Denitrification as well as Its Effect on Phosphorus Release in Sediments of Chinese Shallow Lakes"

_water, doi:10.3390/w11091809_

Round 1
Reviewer 1 Report
Manuscript number 563092
Title:
Coupling between nitrification and denitrification as well as its effect on phosphorus release in sediments of Chinese shallow lakes
The authors studied relationship between nitrification and denitrification and its effect on phosphorus release in shallow lakes with different nutrient gradient.
The proposed topic is worth of investigation, however (i) there were many missing information in the methods; (ii) the statistical sections of the manuscript is unacceptable
These issues should be revised before the manuscript will be considered for publication.
Abstract
Abbreviations should be deleted and introduced later in the Methods section. Abstract should be concise and include aims, short description of results and the main conclusions.
Introduction
There are too many abbreviations in the text that makes it difficult to understand
Materials and methods
Study sites and sample collection
Please delete markings of sampling sites from the text and add them to the caption of Figure 1
There are too many abbreviations in the text that makes it difficult to understand
Samples were taken in May, what about the weather conditions, usually values of TSI are calculated for the summer period, it is extremely important in highly eutrophic and hypertrophic lakes. In summer air temperature is the highest and the processes of P release from sediments are accelerated mostly as a result of low DO concentrations and high biomass of phytoplankton (cyanobacteria prevailed).
How many samples were taken on each site, only one per site and sites were used as replicates to statistical analysis? Please explain.
Statistical analysis
What was the reason of using Pearson test, if the data were not verified for the normal distribution, if not it is not possible to use parametric tests.
Results
Relationship among nutrient species, functional genes and potential rate
From the text, Table2 and Fig.8a-d it follows that Pearson correlation coefficients were calculates for all the data, if so it is not possible to compare lakes of different nutrient levels.
Correlations should be calculated separately for each of studied lake, but if number of replicated is equal the number of sites (n=5) it is not enough to obtain reliable results.
Please clarify number of replicates (n) for each lake and it also in all figures and tables captions
Discussion
Page 11, lines 210-211
„…The trophic state of lakes determines the microbial activity and functional gene abundance…”
I think that prsenetation of results and statistical methods used in the manuscript do not qualify fot the statement.
Discussion should be substantially revised after changes in statistical analyzes
There are too many abbreviations in the whole text that makes it difficult to understand.

Author Response
Response to Reviewer #1:
Title:
Coupling between nitrification and denitrification as well as its effect on phosphorus release in sediments of Chinese shallow lakes
The authors studied relationship between nitrification and denitrification and its effect on phosphorus release in shallow lakes with different nutrient gradient.
The proposed topic is worth of investigation, however (i) there were many missing information in the methods; (ii) the statistical sections of the manuscript is unacceptable
These issues should be revised before the manuscript will be considered for publication.
Thank you very much for your suggestion. We added information in the methods and the statistical section.
Abstract
Abbreviations should be deleted and introduced later in the Methods section. Abstract should be concise and include aims, short description of results and the main conclusions.
Thank you very much for your suggestion. We deleted the abbreviations here and revised the abstract according to your suggestions as follows:
The coupling of nitrification and denitrification has attracted wide attention for this process played an important role in mitigating eutrophication in aquatic ecosystems, while limited information about the mechanism was known. In order to study the coupling relationship between nitrification and denitrification as well as its effect on phosphorus release, three shallow lakes with different nutrient gradient in Wuhan city were selected to analyze the nutrient level and functional gene abundance and potential rate involved in nitrification and denitrification. The statistical analysis showed that, trophic level was positively related with copy numbers of function gene of nitrosomonas and denitrifiers as well as potential nitrification rate and potential denitrification rate. Concentration of different form of phosphorus showed a positive correlation with number of nitrosomonas and denitrifiers as well as potential nitrification rate and potential denitrification rate. The number of function gene for nitrosomonas exhibited positive linear correlations not only with function gene of denitrifiers but also with potential denirification rate. In turn, potential nitrification rate was significantly and positively related with function gene of nitrous oxide reductase. In conclusion, lake eutrophication aggravation (especially the increase of phosphorus concentration ) promoted the processes of nitrification and denitrification by shaping their functional genes. And the close coupling between nitrification and denitrification was mediated by nitrifier denitrification. Strong nitrification-denitrification fuelled the nitrogen removal from the system, and also accelerated the phosphorus release due to anaerobic state caused by organic matter decomposition and nitrification. During this process, dissolved organic nitrogen was also released into water column in quantity, which was in favor in balancing the nitrogen and phosphorus ratio.
Introduction
There are too many abbreviations in the text that makes it difficult to understand
Thank you very much for your suggestion. We deleted abbreviations in the introduction section and introduced later in the methods section.
Materials and methods
Study sites and sample collection
Please delete markings of sampling sites from the text and add them to the caption of Figure 1
There are too many abbreviations in the text that makes it difficult to understand
Thank you very much for your suggestion. We deleted the markings of sampling sites from the text and add them to the caption of Figure 1 according to your suggestion. Also, we added introduction of the abbreviations here and deleted some unnecessary abbreviations.
Samples were taken in May, what about the weather conditions, usually values of TSI are calculated for the summer period, it is extremely important in highly eutrophic and hypertrophic lakes. In summer air temperature is the highest and the processes of P release from sediments are accelerated mostly as a result of low DO concentrations and high biomass of phytoplankton (cyanobacteria prevailed).
Thank you very much for your professional advice, the day of sampling was sunny and the air temperature was about 28 centigrade, water temperature was about 27 centigrade. We added these information in the text.
How many samples were taken on each site, only one per site and sites were used as replicates to statistical analysis? Please explain.
Thank you very much for your question. Only one sample was taken on each site and sites were used as independent to statistical analysis, because the nutritional status may be of significant differences between sampling sites even in the same lake. For example the TSI of HG1 was 66.69 while for HG5, the TSI was 56.66.
Statistical analysis
What was the reason of using Pearson test, if the data were not verified for the normal distribution, if not it is not possible to use parametric tests.
Thank you very much for your question. Data used for correlation analysis were tested before analysis. And all the data were logged before analysis to make sure that the data obey normal distribution. This transformation was reflected in the graphs, but not in table 2, so we add explanations in table 3.
Results
Relationship among nutrient species, functional genes and potential rate
From the text, Table2 and Fig.8a-d it follows that Pearson correlation coefficients were calculates for all the data, if so it is not possible to compare lakes of different nutrient levels.
Correlations should be calculated separately for each of studied lake, but if number of replicated is equal the number of sites (n=5) it is not enough to obtain reliable results.
Please clarify number of replicates (n) for each lake and it also in all figures and tables captions
Thank you very much for your question. We changed “compare lakes of different nutrient levels” into “compare nutrient level between different sampling sites or with other publications” and according to another reviewer’s advise, we combined results and discussion as one section. One sample was taken on each site and sites were used as independent to statistical analysis, because the nutritional status may be of significant differences between sampling sites even in the same lake, so the replicated number of sites was 16. And we added n=16 in table 3 and Figure 8-10.
Discussion
Page 11, lines 210-211
„…The trophic state of lakes determines the microbial activity and functional gene abundance…”
I think that prsenetation of results and statistical methods used in the manuscript do not qualify fot the statement.
Thank you very much for your suggestion. We combined results and discussion together and rewrite this part as follows:
In this study, we observed a close relationship between nutritional status of lakes and the abundance as well as activity of nitrosomonas and denitrifiers (Table 3, Fig. 3 and Fig. 4). These results indicate that the trophic state of lakes determines the microbial activity and functional gene abundance.
Discussion should be substantially revised after changes in statistical analyzes
Thank you very much for your suggestion. Different sites are considered to be independent sampling units, the data were transformed before analysis and normality test was carried out before analysis. So we followed the previous analysis method here.
There are too many abbreviations in the whole text that makes it difficult to understand.
Thank you very much for your suggestion. We deleted some unnecessary abbreviations in the text to make it easier to read, and we added a list of abbreviations at the end of the article.

Reviewer 2 Report
The results of on-time, but very detailed study on nutrients content in water and sediments as well as microbial composition by means of functional gene abundance in three shallow lakes were presented in manuscript. The aim was to determine the coupling relationship between nitrification and denitrification, and finally also its impact on phosphorus release. The results indicate high microbial activity in shallow lakes, related to the process of eutrophication. This was already know, while authors increased the knowledge on nutrient transformations in sediments, especially in the matter of the nitrifier denitrification. Thus, I consider this research very important in understanding the processes behind the eutrophication as well as in possibilities of its mitigation.
My most concern about the data presented in manuscript relates to the assessment of trophic state of studied lakes (lines 67-70). According to Carlson’s TSI:
- Lake Houguan – not oligotrophic according, rather eutrophic
- Lake Tangxun – not middle-eutrophic, but strongly hypertrophic
- Lake Qingling, southern part – not light-eutrophic, but light-hypertrophic
- Lake Qingling, northern part – strongly hypertophic
I also have some minor comments:
line 71 – explain abbreviation EEA in brackets, similarly as PDR, PNR etc.
line 90 – reference shall be as number in []
line 91 – ‘and absorbance’ – what does it mean in this statement?
line 157, Fig. 2 – DOP is presented, while DTP is mentioned in methods – why?
Fig.4 and 6 – bars are hardly to recognize
Fig. 8-10 – legends are too small to read

Author Response
Response to Reviewer #2:
The results of on-time, but very detailed study on nutrients content in water and sediments as well as microbial composition by means of functional gene abundance in three shallow lakes were presented in manuscript. The aim was to determine the coupling relationship between nitrification and denitrification, and finally also its impact on phosphorus release. The results indicate high microbial activity in shallow lakes, related to the process of eutrophication. This was already know, while authors increased the knowledge on nutrient transformations in sediments, especially in the matter of the nitrifier denitrification. Thus, I consider this research very important in understanding the processes behind the eutrophication as well as in possibilities of its mitigation.
Thank you very much for your positive comment.
My most concern about the data presented in manuscript relates to the assessment of trophic state of studied lakes (lines 67-70). According to Carlson’s TSI:
- Lake Houguan – not oligotrophic according, rather eutrophic
- Lake Tangxun – not middle-eutrophic, but strongly hypertrophic
- Lake Qingling, southern part – not light-eutrophic, but light-hypertrophic
- Lake Qingling, northern part – strongly hypertophic
Thank you very much for your professional explanation, it was our mistake and we revised the description in the text.
I also have some minor comments:
line 71 – explain abbreviation EEA in brackets, similarly as PDR, PNR etc.
Thank you very much for your suggestion. The meaning and testing methods were detailed introduced in 2.2 and 2.3, so we deleted this abbreviations here and added detailed explanation in 2.3. We added a list of abbreviations at the end of the article.
line 90 – reference shall be as number in []
Thank you very much for your suggestion. We revised as you suggested.
line 91 – ‘and absorbance’ – what does it mean in this statement?
Thank you very much, we made a mistake here and we deleted ‘and absobance’ in the sentence.
line 157, Fig. 2 – DOP is presented, while DTP is mentioned in methods – why?
Thank you very much for your suggestion. We have adjusted the figure 2, changed DOP into DTP.
Fig.4 and 6 – bars are hardly to recognize
Thank you very much for your suggestion. We deleted NO2--N in Fig. 4(for this data was too small) and transferred Fig.6 into a table to make it clearer.
Fig. 8-10 – legends are too small to read
Thank you very much for your suggestion. We revised the legends as you suggested.

Reviewer 3 Report
This paper shows Coupling between nitrification and denitrification as well as its effect on phosphorus release in sediments of Chinese shallow lakes. The paper is interesting but, in the current form is not ready for the publication and needs substantial changes. Therefore, I suggest major revision to see the author’s improvement especially in language and paper structure.
List of the abbreviations should be added.
Abstract: it should be revised. The sentences are too long and not connected to each other. The novelty is not clear. It should start with the problem statement and then the aim of the research. A big part of the abstract is results and used methodology is not included.
Introduction: The provided information is good and relevant while the reference should be updated to recent. Also, this part also has a language problem like lines 36-37, line 48 and many more.
Line 56: why N removal and P elimination? Both can be removal.
Lines 57-60 is copied from the abstract which is too long and not clear.
Line 67: Did the authors used three lakes or four lakes for the sampling?
Line 78: add some detail for Secchi disk method and device. Also, needs a reference.
Table 1: how much was the water temperature during sampling?
The explanation for figures 2 to 7 is not enough and critical. Also, don’t have any comparison with published studies. I suggest merging these figures in one.
Line 241: don’t use “will” in the paper.
The discussion part is written separately and it is difficult to refer to all the figures one by one. I suggest merging this section with results and then create a short section to connect the obtained results from figures 2-7 and 8-10 in a single section.
The conclusion also is not strong and needs to be improved by more quantitative results.
References should be updated to the recent and it can be reduced to fewer references.
Author Response
Response to Reviewer #3:
This paper shows Coupling between nitrification and denitrification as well as its effect on phosphorus release in sediments of Chinese shallow lakes. The paper is interesting but, in the current form is not ready for the publication and needs substantial changes. Therefore, I suggest major revision to see the author’s improvement especially in language and paper structure.
List of the abbreviations should be added.
Thank you very much for your suggestion. We seek for someone who is a native English writer to correct the English expression through English Language Editing and changed a lot on the paper structure as you suggested. And we added a list of the abbreviations at the end of the article.
Abstract: it should be revised. The sentences are too long and not connected to each other. The novelty is not clear. It should start with the problem statement and then the aim of the research. A big part of the abstract is results and used methodology is not included.
Thank you very much for your suggestion. We revised and almost re-wrote the abstract, highlighted the important points and deleted irrelevant results, The “Abstract” part was re-wrote as follows:
Abstract: The coupling of nitrification and denitrification has attracted wide attention for this process played an important role in mitigating eutrophication in aquatic ecosystems, while limited information about the mechanism was known. In order to study the coupling relationship between nitrification and denitrification as well as its effect on phosphorus release, three shallow lakes with different nutrient gradient in Wuhan city were selected to analyze the nutrient level and functional gene abundance and potential rate involved in nitrification and denitrification. The statistical analysis showed that, trophic level was positively related with copy numbers of function gene of nitrosomonas and denitrifiers as well as potential nitrification rate and potential denitrification rate. Concentration of different form of phosphorus showed a positive correlation with number of nitrosomonas and denitrifiers as well as potential nitrification rate and potential denitrification rate. The number of function gene for nitrosomonas exhibited positive linear correlations not only with function gene of denitrifiers but also with potential denirification rate. In turn, potential nitrification rate was significantly and positively related with function gene of nitrous oxide reductase. In conclusion, lake eutrophication aggravation (especially the increase of phosphorus concentration ) promoted the processes of nitrification and denitrification by shaping their functional genes. And the close coupling between nitrification and denitrification was mediated by nitrifier denitrification. Strong nitrification-denitrification fuelled the nitrogen removal from the system, and also accelerated the phosphorus release due to anaerobic state caused by organic matter decomposition and nitrification. During this process, dissolved organic nitrogen was also released into water column in quantity, which was in favor in balancing the nitrogen and phosphorus ratio.
Introduction: The provided information is good and relevant while the reference should be updated to recent. Also, this part also has a language problem like lines 36-37, line 48 and many more.
Thank you very much for your suggestion. We revised this part and update reference, please see the “introduction” part.
Line 56: why N removal and P elimination? Both can be removal.
Thank you very much for your question. “P elimination” may be inappropriate here, in fact, we mean P release from sediments. Enrichment of organic carbon and nitrogen can result in P release through the formation of anaerobic status caused by organic matter decomposition . We changed “elimination” into “release” here.
Lines 57-60 is copied from the abstract which is too long and not clear.
Thank you very much for your suggestion. We revised this sentence as follows:
It is mainly because the anaerobic state accelerated the P release.
Line 67: Did the authors used three lakes or four lakes for the sampling?
Thank you very much for your question. Lake QingLing was separated into two parts and the two section were of great difference so we said “we used four representative lakes or zones”.
Line 78: add some detail for Secchi disk method and device. Also, needs a reference.
Thank you very much for your suggestion. We added some detail for Secchi disk method and added a reference as you suggested.
Table 1: how much was the water temperature during sampling?
Thank you very much for your question. The water temperature was about 27 degrees with minor variations, so we did not list this data in table 1 and we added information in the method section.
The explanation for figures 2 to 7 is not enough and critical. Also, don’t have any comparison with published studies. I suggest merging these figures in one.
Thank you very much for your suggestion. We merged figure 2, 3, 4, 5 and 7 into one figure, we transfer figure 6 into a table for the bars of figure 6 are hardly to recognize. And we added explanation for figure 2 and compared our results with others.
Line 241: don’t use “will” in the paper.
Thank you very much for your suggestion. We revised “will” into “would”.
The discussion part is written separately and it is difficult to refer to all the figures one by one. I suggest merging this section with results and then create a short section to connect the obtained results from figures 2-7 and 8-10 in a single section.
Thank you very much for your suggestion. We merged the discussion section and the results section into one as you suggested.
The conclusion also is not strong and needs to be improved by more quantitative results.
Thank you very much for your suggestion. We combined the results and the discussion together and re-wrote this part. Please see the “Results and Discussion” section.
References should be updated to the recent and it can be reduced to fewer references.
Thank you very much for your suggestion. We updated and reduced the references as you suggested.

Round 2
Reviewer 1 Report
Manuscript number 563092
Comments to revised version
Title:
Coupling between nitrification and denitrification as well as its effect on phosphorus release in sediments of Chinese shallow lakes
The Authors revised the manuscript referring to most of the comments contained in the review.
Still the statistical analysis and explanations are not acceptable.
If there were only 5 sites per lake and they differred in TSI values (e.g. TSI of HG1 amounted 66.69 and for HG5 - 56.66). It is impossible to consider them as independent replicates, there should be more replicates for the site to verify the real TSI value. If so, results of statistical analysis are not reliable.
Moreover the Authors changed “compare lakes of different nutrient levels” into “compare nutrient level between different sampling sites or with other publications”
It does not make any sense, what was the reason to compare sites between lakes or with other publications?
Still the numer of replicates is too low to obtain the relaible results
Correlations should be calculated separately for each of studied lake with regard to its trophic status.

Author Response
Response to Reviewer #1:
Title:
Coupling between nitrification and denitrification as well as its effect on phosphorus release in sediments of Chinese shallow lakes
The Authors revised the manuscript referring to most of the comments contained in the review.
Still the statistical analysis and explanations are not acceptable.
If there were only 5 sites per lake and they differred in TSI values (e.g. TSI of HG1 amounted 66.69 and for HG5 - 56.66). It is impossible to consider them as independent replicates, there should be more replicates for the site to verify the real TSI value. If so, results of statistical analysis are not reliable.
Moreover the Authors changed “compare lakes of different nutrient levels” into “compare nutrient level between different sampling sites or with other publications”
It does not make any sense, what was the reason to compare sites between lakes or with other publications?
Still the numer of replicates is too low to obtain the relaible results
Correlations should be calculated separately for each of studied lake with regard to its trophic status.
Thank you very much for your professional suggestions. Your understanding and opinion is right. If we want to compare the difference of measured parameters in different lakes, five sites for per lake for one time sampling is not enough to get the reliable results. Also, the related statistical analysis and correlations analysis needed more sites and data to support. However, in this study, the comparison of parameters in different lake is not our key point. The purpose of this study is not to compare the difference of parameters among all studied lakes, but try to seek for a general principle from numerous lakes with different environmental conditions. In this case, the correlations analysis with more different lakes and different nutrient level is necessary for us to get the common pattern, regardless of oligotrophic or eutrophic. Hence, through the correlation analysis including all our studied lakes data, we found some common rule, such as the relationship between nutrient level and abundance of nitrifier and denitrifiers as well as potential rate, also we found the relationship between potential rate and nutrient release, and so on.
Thus, the comparison among different sites is not necessary and does not make any sense as you said, we deleted these description about difference among different sites. Actually, we started the lake survey three years before the sampling time in this article, involving 38 lakes and 124 sampling sites. After that, we selected typical representative lakes or zones (including the lakes of this article) with different nutrient gradients to sample every month to conduct in-depth research on the mechanism of N and P Cycle. The specific value of TSI might change over time but the tendency remained unchanged and the TSI data in this paper were close to the months before and after.
That is our explanation, and I hope you can satisfy it. Thank you again.

Reviewer 3 Report
The revised version of the manuscript (water-563092) looks better but still needs too many changes. It has several grammatical errors while some of the comments didn’t address properly. I suggest major revision.
Abstract: not much changed. Still has long sentences with language problems. Authors should revise it completely not just simply add some sentences.
Introduction: again lines 63-66 are copied from the abstract. I wonder how the authors didn’t consider the comments like this. Don’t start sentences with “And”. This part should be expanded to more information related to the aim of the study.
Material and methods: what is Dissolved organic nitrogen? Are the authors didn’t see this phrase in other papers too?
Results and discussion: this part needs to be restructured. Figures 3 to 5 should be separated, and the related text should be moved close to each figure. Now is confusing for the readers.
Conclusion: why the authors didn’t consider the previous comments? The conclusion also is not strong and needs to be improved by more quantitative results.
References: what is reference number 17? What is the relation of reference number 35 to this study? The majority of the added references are from a specific country which is not recommended.
Author Response
Response to Reviewer #3:
The revised version of the manuscript (water-563092) looks better but still needs too many changes. It has several grammatical errors while some of the comments didn’t address properly. I suggest major revision.
Thank you very much for your suggestion, the comments are helpful for the improvement of our manuscript. We have revised the manuscript accordingly and we chose a professional English editing service provided by MDPI service to help us with the language issues.
Abstract: not much changed. Still has long sentences with language problems. Authors should revise it completely not just simply add some sentences.
Thank you very much for your suggestion. I seriously considered your suggestion and revised the Abstract by a large degree, including separate the long sentence, incorrect grammar, and confused sentences. Please see the follows:
The coupling of nitrification and denitrification has attracted wide attention since it plays an important role in mitigating eutrophication in aquatic ecosystems. While, the underlying mechanism is largely unknown. In order to study the coupling relationship between nitrification and denitrification as well as its effect on phosphorus release, nutrient levels, functional gene abundance and potential rates involved in nitrification and denitrification were analyzed in three shallow urban lakes with different nutrient status. Trophic level was found positively related to not only copy numbers of functional genes of nitrosomonas and denitrifiers, but also the potential nitrification and denitrification rates. In addition, the concentrations of different forms of phosphorus showed a positive correlation with the number of nitrosomonas and denitrifiers as well as potential nitrification and denitrification rates. Furthermore, the number of functional genes of nitrosomonas exhibited positive linear correlations with functional genes and rate of denitrification. These facts suggested that increase in phosphorus concentration might promote the coupling of nitrification and denitrification by increasing their functional genes. Strong nitrification–denitrification fuelled the nitrogen removal from the system, and accelerated the phosphorus release due to the anaerobic state caused by organic matter decomposition and nitrification. Moreover, dissolved organic nitrogen was also released into the water column during this process, which was favorable for balancing the nitrogen and phosphorus ratio. In conclusion, the close coupling between nitrification and denitrification mediated by nitrifier denitrification made an important effect on the cycling mode of nitrogen and phosphorus.
Introduction: again lines 63-66 are copied from the abstract. I wonder how the authors didn’t consider the comments like this. Don’t start sentences with “And”. This part should be expanded to more information related to the aim of the study.
Thank you very much for your suggestion. We added information in this part, deleted the “and” if the first word is “and” in a sentence. We revised the sentence from line 63 to 66 as follows:
In this study, three shallow lakes with different nutrient gradients in Wuhan city were selected to analyze nutrient levels, functional genes abundance and potential rates involved in nitrification and denitrification. It was supposed to find out the relationship among nitrogen and phosphorus level, nitrifying and denitrifying bacteria abundance and their potential rate. We hope to verify this hypothesis that the abundance of nitrifying and denitrifying bacteria determined the potential rate of nitrification and denitrification as well as their coupling relationship, which further stimulated the nitrate removal and phosphorus release.
Material and methods: what is Dissolved organic nitrogen? Are the authors didn’t see this phrase in other papers too?
Thank you very much for your professional question. Actually, dissolved organic nitrogen has not received more attention, especially in freshwater ecosystems. At the earliest, dissolved organic nitrogen was focused on the ocean due to its significance on the hydrolysis to produce available inorganic nitrogen. Dissolved organic nitrogen was mainly included some amino acid with small weight. Now in lakes and rivers, scientists began to believe that DON could contribute to water body eutrophication in a large degree, and in some lakes, DON could contribute the above 50% TN level. Please refer to some reference as follows:
Riggsbee, J. A. , Orr, C. H. , Leech, D. M. , Doyle, M. W. , & Wetzel, R. G. . (2015). Suspended sediments in river ecosystems: photochemical sources of dissolved organic carbon, dissolved organic nitrogen, and adsorptive removal of dissolved iron. Journal of Geophysical Research Biogeosciences, 113(G3), 194-198.
Shammon, T. M. , & Hartnoll, R. G. . (2002). The winter and summer partitioning of dissolved nitrogen and phosphorus. observations across the irish sea during 1997 and 1998. Hydrobiologia, 475-476(1), 173-184.
Results and discussion: this part needs to be restructured. Figures 3 to 5 should be separated, and the related text should be moved close to each figure. Now is confusing for the readers.
Thank you very much for your suggestions. As your indication, we restructured the Results and discussion part to make the relative text and figure close and to be convenient for the reader to follow the figure and text. Also, we revised the order of relative text to make the MS more self-consistent and logical.
Conclusion: why the authors didn’t consider the previous comments? The conclusion also is not strong and needs to be improved by more quantitative results.
Thank you very much for your suggestions. As you indicated, we revised the Conclusion part as follows:
“Taken together, compared to eutrophic Lake Houguan, nutrients enrichment (P in particular) in sediments of strongly hypertophic North of Lake Qingling greatly increased nitrifying and denitrifying microbial abundance by an order of magnitude, further accelerating the potential rate of nitrification and denitrification by 6.9 and 3.5 fold respectively, especially the coupling of these two processes mediating by nitrifier denitrification. Strong nitrification-denitrification in sediments of North of Lake Qingling fuelled the nitrogen removal from the system in terms of 21.9% and 27.0% lower ammonium and nitrate level than that of LakeHouguan. Meantime, this coupling also accelerated the P release (1000 times higher SRP value in water column of North of Lake Qingling, comapred to Lake Houguan) due to anaerobic state caused by organic matter decomposition and nitrification. During this process, DON was also released into water column in quantity, which was in favor of balancing the N and P ratio.”
References: what is reference number 17? What is the relation of reference number 35 to this study? The majority of the added references are from a specific country which is not recommended.
Thank you very much for your careful suggestion. It is our negligence. We have supplemented the reference number 17, and deleted the reference number 35. At the same time, we updated the related and valuable references and deleted some specific or regional reference.

Round 3
Reviewer 1 Report
Manuscript number 563092
Final comments
Title:
Coupling between nitrification and denitrification as well as its effect on phosphorus release in sediments of Chinese shallow lakes
The Authors revised the manuscript referring to the review comments. The manuscript can be published in present form.

Author Response
Thank you very much for your suggestions, the previous comments are really helpful for the improvement of our manuscript.

Reviewer 3 Report
I reviewed the second revision of (water-563092) based on the previous comments given by different reviewers. I can see a huge change by the authors in term of language and structure of the paper. I suggest a minor revision.
Some of the figures have a problem. The inside writing text is overlapping. example figure 5. Figures 2b and 2 c are unclear. the colors are almost the same and can not be differentiated. Text in the vertical axis has overlapping. Figure 4 also has some problems. Figure 3 d has two different details inside the box while figures 4 ( e and f ) don't have any. The results and discussion section is separately written. You can add the heading discussion after Figure 5. the deleted reference Liu et al 2018 is not suggested. You can keep as it is related and recent. Added reference Carrasco 2004 is not suggested. You should balance between old and recent. The majority needs to be recent and surely relevant.
Author Response
I reviewed the second revision of (water-563092) based on the previous comments given by different reviewers. I can see a huge change by the authors in term of language and structure of the paper. I suggest a minor revision.
Some of the figures have a problem. The inside writing text is overlapping. example figure 5. Figures 2b and 2 c are unclear. the colors are almost the same and can not be differentiated. Text in the vertical axis has overlapping. Figure 4 also has some problems. Figure 3 d has two different details inside the box while figures 4 ( e and f ) don't have any. The results and discussion section is separately written. You can add the heading discussion after Figure 5. the deleted reference Liu et al 2018 is not suggested. You can keep as it is related and recent. Added reference Carrasco 2004 is not suggested. You should balance between old and recent. The majority needs to be recent and surely relevant.
Thank you very much for your careful suggestion, the comments are really helpful for the improvement of our manuscript. We adjusted the inside writing text in the figures, blue and red were used in figure 2b and 2c to increase image recognition. Both figure 3e and 3f reflected only one set of correlation, so there was only one detail inside the box and we adjusted the position to the lower right corner of the picture to keep it in line with other figures.
We added the heading discussion after figure 5 as you suggested.
We added reference Liu et al 2018 and deleted Carrasco 2004 as you suggested.
Once again, thank you very much for your suggestions, it is really helpful for the improvement of our manuscript.
